# RISK-AWARE ROBUST GRAPH NETWORK EXPLANATION

## ABSTRACT

Post-hoc explanation methods for Graph Neural Networks (GNNs) are increasingly used to reveal which substructures influence a model's prediction. However, recent studies show that such explanations are often brittle—small changes to the input graph can lead to drastically different explanations. This instability challenges their reliability in critical downstream tasks such as auditing, debugging, or human-in-the-loop decision making.

In this work, we introduce GrA, a risk-aware explanation trimming method that enhances the robustness of GNN explanations via a post-hoc, model-agnostic process. GrA identifies unstable edges using gradient-based sensitivity analysis and quantifies their volatility via Conditional Value-at-Risk (CVaR), a tail-aware risk measure. By removing high-risk edges, GrA produces a robust surrogate graph that retains explanatory fidelity while significantly reducing sensitivity to structural perturbations.

GrA requires no modification to the underlying GNN or explanation model and can be seamlessly applied to any gradient-accessible explainer. Across both synthetic and real-world graph classification benchmarks, and under various adversarial perturbation settings, GrA consistently improves explanation stability without compromising fidelity or predictive accuracy.

## 1 INTRODUCTION

Graph Neural Networks (GNNs) have become a standard tool for learning over graphs, attaining state-of-the-art results in molecular property prediction, fraud detection, recommendation, and more. As these models are increasingly deployed in high-stakes settings, the need for *faithful and robust* explanations is critical—for user trust, regulatory compliance, debugging, and human-in-the-loop decision-making. Among many approaches, *post-hoc* explainers such as GNNEXPLAINER Ying et al. (2019) and PGEXPLAINER Luo et al. (2020) are widely used to identify subgraph structures that support a trained model's decision.

**Problem.** Despite their popularity, current explainers are *brittle*: small graph perturbations (e.g., adding or deleting a handful of edges) can produce drastically different explanations (Figure. 1), even when the predicted class is unchanged. This instability undermines the reliability of explanations in dynamic or adversarial environments and complicates auditing. Existing defenses typically (i) modify training to induce more stable attributions, or (ii) provide certified guarantees that are often limited to small graphs. Neither perspective treats instability as an *intrinsic property of the input graph* that can be managed post hoc, independent of the predictor.

We consider a trained GNN $f$ for node classification and a post-hoc explainer $g$. Given a graph $G = (\mathcal{V}, \mathcal{E})$ with adjacency $A$ and features $X$, the explainer returns a *soft* edge-importance vector $M^{(v)} = g(G, f; v) \in [0,1]^m$ for a target node $v$, where $m = |\mathcal{E}|$. We ask: *which edges in $G$ are responsible for the observed instability of $M^{(v)}$ under small structural changes?* We will formalize an *edge risk* that quantifies how sensitive $M^{(v)}$ is to infinitesimal changes of each edge and show that risks exhibit a long-tail pattern (§2, §3).

**Approach: risk-aware trimming.** We propose **GrA** (Graph risk-Aware trimming), a training-free, model-agnostic module that improves explanation stability by *trimming* edges identified as

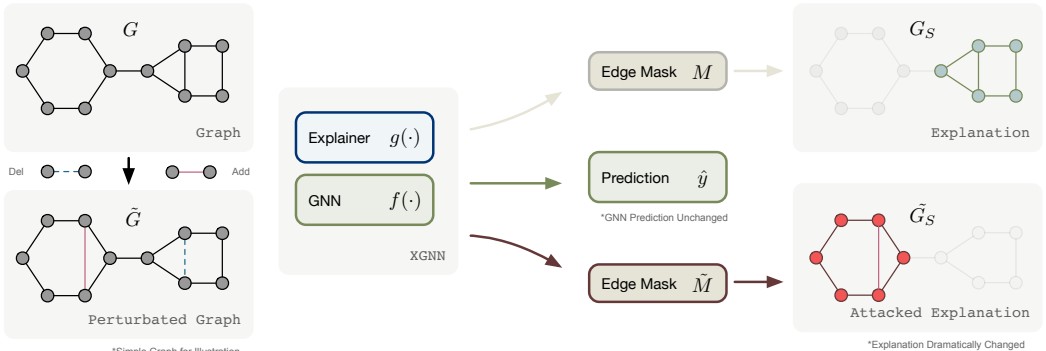

Figure 1: **Motivation: explanation instability under small perturbations.** For a target node $v$, a post-hoc explainer $g$ outputs an edge-importance vector $M^{(v)} \in [0,1]^m$ aligned with the edges $\mathcal{E}$ of the input graph $G = (\mathcal{V}, \mathcal{E})$ (see §2). Minor structural edits to $G$ (few edge additions/deletions) can drastically alter $M^{(v)}$ while leaving the model prediction unchanged.

*high-risk*. Concretely, we introduce continuous edge gates $z \in \mathbb{R}^m$ to differentiate through the explainer and define the risk of edge $e$ as the $\ell_2$ sensitivity $\left\| \partial M^{(v)} / \partial z_e \right\|_2$. Motivated by risk management, we select edges using the *Conditional Value-at-Risk* (CVaR) of the empirical risk distribution: CVaR focuses on the severity of the worst $(1-\alpha)$ fraction of edges, enabling a *tail-aware* trimming rule with a minimum retention ratio $\delta$. An optional importance-preserving trade-off further protects highly informative edges while suppressing volatile ones (§3).

**Why this design.** (i) *Post-hoc*: keeps the predictor $f$ unchanged and applies across datasets/architectures. (ii) *Model-agnostic*: works with any differentiable explainer $g$ (GCN is our default backbone for clarity, but GrA is not tied to GCN). (iii) *Attacker-agnostic*: targets the root cause—tail instability in the input graph—and improves robustness against diverse perturbation-based attacks that keep predictions invariant.

**Contributions.**

- **Edge risk.** We introduce a formal notion of *explanation risk* per edge, defined as the gradient sensitivity of the explainer's soft mask to continuous edge gates; empirically, risks are long-tailed.

- **Tail-risk trimming.** We develop a CVaR-based, closed-form trimming rule with a retention constraint and an importance-preserving trade-off, yielding robust surrogate graphs and more stable explanations.

- **Comprehensive evaluation.** Across synthetic and real-world benchmarks, explainers (e.g., GNNEXPLAINER, PGEXPLAINER), and perturbation-based attacks, GrA consistently improves stability (IoU/consistency, rank correlation) while maintaining fidelity and predictive accuracy, with modest overhead.

We focus on node-level post-hoc explanations (clear evaluation and notation), instantiate $f$ with a GCN by default, and evaluate under perturbation budgets that preserve the predicted label. The methodology is broadly applicable and requires no retraining of $f$ or $g$.

## 2 PRELIMINARIES

### 2.1 GRAPHS AND GNNS

Let $G = (\mathcal{V}, \mathcal{E})$ be a graph with $n = |\mathcal{V}|$ nodes and $m = |\mathcal{E}|$ edges, described by a binary adjacency matrix $A \in \{0,1\}^{n \times n}$ and a node feature matrix $X \in \mathbb{R}^{n \times d}$. We consider the task of **node classification**, where a trained GNN model $f$ maps the graph data $(A, X)$ to a vector of logits $f_v(A, X) \in \mathbb{R}^C$ for a target node $v \in \mathcal{V}$, predicting its class label. While we focus on GCNs as a representative backbone, our framework is applicable to general message-passing architectures.

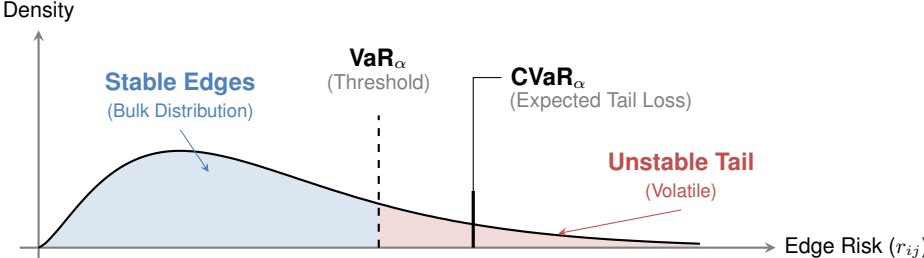

Figure 2: **Edge Risk Distribution and Measures.** An illustration of the long-tail distribution observed in GNN explanation sensitivities (edge risks). Most edges are stable (bulk), while a minority exhibit high volatility (tail). VaR marks a threshold, while CVaR quantifies the expected severity of this tail, serving as a robust objective for trimming unstable structures. Although the high-risk edges (red region) are few in number, they exhibit disproportionately high volatility. Consequently, these extreme outliers are the primary drivers of instability and the main targets for removal in our GrA framework.

## 2.2 POST-HOC EXPLANATIONS: SCOPE AND DEFINITION

**Taxonomy and Focus.** GNN explainability methods typically fall into two categories: *intrinsic* methods, which modify the model architecture to be self-interpretable (e.g., via attention weights), and *post-hoc* methods, which interpret a fixed, pre-trained model. In this work, we focus on **post-hoc explainers**. This choice is motivated by their *model-agnostic* nature: they can be applied to any trained GNN without sacrificing predictive performance or requiring retraining, making them critical for auditing deployed systems.

**Formulation.** Given a trained model $f$ and an input graph $G$, a post-hoc explainer $g$ seeks to identify a substructure that is most influential for the prediction at target node $v$. Formally, the explainer produces an *edge-importance mask* $M^{(v)} = g(G, f; v) \in [0, 1]^m$, where each entry $M_{ij}^{(v)}$ represents the relevance of edge $e_{ij}$. Prominent examples include GNNEXPLAINER (Ying et al., 2019), which optimizes a mask to maximize mutual information, and PGEXPLAINER (Luo et al., 2020), which learns a parameterized edge-scoring network. The final explanation is typically a subgraph $G_S$ induced by edges with importance scores exceeding a threshold.

## 2.3 RISK MEASURES: FROM VAR TO CVAR

To quantify the instability of explanations, we adopt concepts from financial risk management. Let $R$ be a real-valued random variable representing a loss or risk profile (e.g., the sensitivity of edges).

**Value-at-Risk (VaR).** For a confidence level $\alpha \in (0, 1)$, the *Value-at-Risk* is simply the $\alpha$-quantile of the distribution:

$$\text{VaR}_\alpha(R) = \inf\{ t \in \mathbb{R} : \mathbb{P}(R \leq t) \geq \alpha \}. \tag{1}$$

While intuitive, VaR acts as a simple cut-off. As illustrated in Figure 2, it is not a *coherent risk measure* (Artzner et al., 1999) because it disregards the shape and magnitude of the tail distribution beyond the threshold.

**Conditional Value-at-Risk (CVaR).** To account for the magnitude of extreme risks, we employ *Conditional Value-at-Risk* (also known as Expected Shortfall). CVaR measures the expected value of the variable given that it exceeds the VaR threshold:

$$\text{CVaR}_\alpha(R) = \mathbb{E}\big[ R \mid R \geq \text{VaR}_\alpha(R) \big]. \tag{2}$$

CVaR is coherent and convex. As shown in Figure 2, CVaR effectively captures the center of mass of the high-risk tail. In our methodology, we demonstrate that optimizing the CVaR of edge sensitivities—rather than simple thresholding (VaR)—is essential for mitigating the impact of "heavy-tailed" structural outliers.

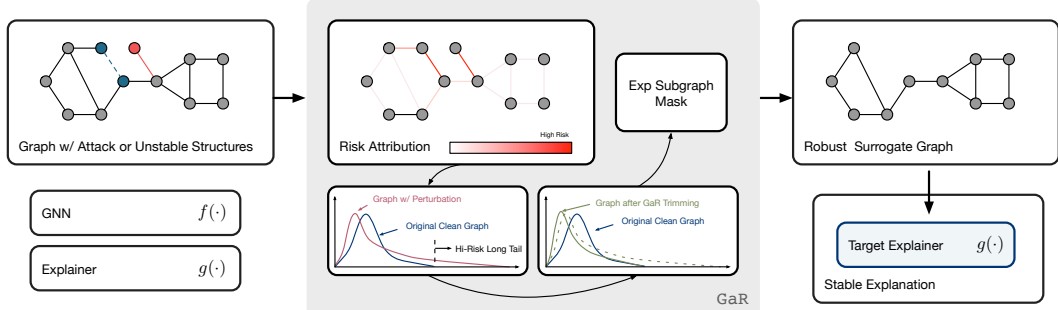

Figure 3: **Overview of the GrA Framework.** (1) **Problem Formulation & Relaxation:** We introduce continuous gate logits $z$ to relax the discrete graph structure, making the explanation mask $M(z)$ differentiable. (2) **Risk Estimation:** We compute Edge Risk $r_{ij}$ as the sensitivity of $M$ to gates $z$ using the efficient **Hutchinson diagonal estimator**. (3) **Tail-Risk Trimming:** Recognizing the **long-tail** nature of edge risks, we use Conditional Value-at-Risk (**CVaR**) to identify edges that contribute disproportionately to instability (considering both frequency and severity). (4) **Robust Output:** High-risk edges are trimmed subject to a retention constraint, yielding a **robust surrogate graph** $\tilde{G}$ for the target explainer.

## 3 METHODOLOGY

We propose **GrA** (Graph Risk-Aware trimming), a post-hoc framework designed to enhance the stability of GNN explanations. GrA operates on the principle that explanation instability is driven by a sparse subset of "volatile" edges. By identifying these edges via gradient sensitivity and managing them through a tail-risk optimization objective, GrA produces a robust surrogate graph $\tilde{G}$ for downstream explanation tasks.

The overall pipeline is illustrated in Figure 3. It consists of three stages: (1) **Risk Modeling**, where we define a differentiable edge risk via continuous relaxation; (2) **Tail-Risk Assessment**, where we employ CVaR to identify high-risk outliers in the long-tail distribution; and (3) **Robust Trimming**, which generates the stable graph using a retention-constrained policy.

### 3.1 MODELING EXPLANATION RISK VIA CONTINUOUS RELAXATION

**Problem Formulation.**    Consider the setup in Sec. 2.1. For a target node $v$, the explainer generates an importance mask. **For brevity, we omit the target index and denote the explanation vector simply as $M \in [0,1]^m$.** Our goal is to quantify the *risk $r_{ij}$* of each edge $e_{ij}$—defined as the sensitivity of the explanation $M$ to the structural presence of that edge.

**Continuous Gating Mechanism.**    Since the graph structure is discrete ($A_{ij} \in \{0, 1\}$), direct differentiation is impossible. We adopt a **Continuous Gating** mechanism for relaxation. For every edge $e_{ij} \in \mathcal{E}$, we introduce a learnable gate logit $z_{ij} \in \mathbb{R}$ and map it to a continuous weight $w_{ij} \in (0, 1)$ via a temperature-scaled sigmoid:

$$w_{ij} = \sigma_T(z_{ij}) = \frac{1}{1 + \exp(-z_{ij}/T)}. \tag{3}$$

We construct a gated adjacency $\tilde{A}(z) = A \odot W$, making the explainer's output a differentiable function of the gates: $M(z) = g(G(\tilde{A}(z), X), f)$. This formulation allows us to treat structural perturbations mathematically as infinitesimal displacements of the continuous variable $z$.

**Edge Risk Definition.**    We define the risk of an edge not by the change in the model's loss, but by the magnitude of the change in the *entire explanation mask*.

**Definition 3.1** (Edge Risk). *The risk $r_{ij}$ of an edge $e_{ij}$ is defined as the $\ell_2$-norm of the gradient of the explanation vector $M(z)$ with respect to the gate variable $z_{ij}$:*

$$r_{ij} \triangleq \left\| \frac{\partial M(z)}{\partial z_{ij}} \right\|_2 = \sqrt{\sum_{k \in \mathcal{E}} \left( \frac{\partial M_k(z)}{\partial z_{ij}} \right)^2}. \tag{4}$$

Physically, $r_{ij}$ quantifies the rate of change of the explanation vector $M$ given an infinitesimal perturbation to edge $e_{ij}$. A high $r_{ij}$ identifies the edge as a "source of sensitivity." To compute Eq. 4 efficiently, we utilize a Hutchinson trace estimator (detailed in Appendix A.3), which approximates the Frobenius norm of the Jacobian with $O(1)$ complexity.

### 3.2 Tail-Risk Assessment: Why CVaR?

Empirical observation reveals that edge risks $\{r_{ij}\}$ follow a **long-tail distribution**: the majority of edges are stable, while a small fraction exhibits extreme volatility. To robustly identify these outliers, we utilize **Conditional Value-at-Risk (CVaR)**.

**CVaR vs. Simple Thresholding.** A naive approach is to prune the top-$K\%$ risky edges, equivalent to using Value-at-Risk ($\text{VaR}_\alpha$) as a threshold. However, VaR is not a coherent risk measure; it only identifies the *frequency* of risk (ranking). In contrast, $\text{CVaR}_\alpha(\mathcal{R})$ captures the *severity* (expected magnitude) of the tail events.

$$\text{CVaR}_\alpha(\mathcal{R}) = \mathbb{E}[r \mid r \geq \text{VaR}_\alpha(\mathcal{R})]. \tag{5}$$

By optimizing the trimmed graph to minimize CVaR, GrA explicitly targets the edges that contribute disproportionately to the aggregate instability. This is mathematically equivalent to minimizing the system's average sensitivity under the worst-case subset scenario.

**Risk-Importance Balancing.** To prevent the removal of semantically critical edges (e.g., edges in a functional motif) that naturally have high gradients, we define a removal score $s_{ij}$ that balances the normalized risk $\hat{r}_{ij}$ against the edge's original importance $\hat{M}_{ij}$:

$$s_{ij} = \lambda_r \cdot \hat{r}_{ij} - \lambda_m \cdot \hat{M}_{ij}. \tag{6}$$

GrA identifies the set of edges $\mathcal{E}_{\text{drop}}$ such that their removal minimizes the tail risk defined by $s_{ij}$, subject to retention constraints.

### 3.3 The GrA Trimming Framework

The framework operates as a "Safety Filter" for GNN explanations. The procedure is summarized in Algorithm 1.

**Retention-Constrained Trimming.** Blindly removing high-risk edges may disconnect the graph. We impose a **Minimum Retention Ratio** $\delta \in (0, 1]$. We select a dynamic threshold $\tau$ based on the $\alpha$-quantile of scores. If the number of remaining edges falls below $\delta|\mathcal{E}|$, we relax $\tau$ to ensure a valid graph structure. The resulting robust graph is $\tilde{G} = (\mathcal{V}, \mathcal{E} \setminus \mathcal{E}_{\text{drop}})$.

**Surrogate Strategy for Black-Box Explainers.** While Definition 3.1 relies on gradients, GrA is not limited to differentiable explainers. For non-differentiable or black-box explainers (e.g., SubgraphX), we employ a **Surrogate Strategy**: 1. Use a lightweight, differentiable surrogate explainer $g_{\text{surr}}$ to estimate the risk map $\{r_{ij}\}$. 2. Generate the trimmed graph $\tilde{G}$ using GrA based on the surrogate risks. 3. Feed the robust graph $\tilde{G}$ into the target black-box explainer. This effectively decouples risk estimation from explanation generation.

### 3.4 Theoretical Stability Insight

We provide a theoretical motivation for why risk trimming enhances stability. We view the explanation process as a map $F : \mathcal{Z} \rightarrow \mathcal{M}$. The stability of this system under perturbation is governed by its **Lipschitz constant** $L$. For any perturbation $\|\Delta z\| \leq \epsilon$, we have $\|\Delta M\| \leq L \cdot \epsilon$.

---

**Algorithm 1** GrA: Graph Risk-Aware Trimming

---

**Require:** Graph $G$, Model $f$, Explainer $g$, Tail $\alpha$, Retention $\delta$, Weights $\lambda$

1: **Step 1: Risk Estimation**
2: Initialize gates $z$; Compute gradients $\nabla_z M$ via Hutchinson estimator.
3: Calculate Edge Risk $r_{ij} \leftarrow \|\nabla_{z_{ij}} M\|_2$.
4: **Step 2: Score Calculation**
5: Normalize risks $\hat{r}$ and original importance $\hat{M}$.
6: Compute scores $s_{ij} \leftarrow \lambda_r \hat{r}_{ij} - \lambda_m \hat{M}_{ij}$.
7: **Step 3: CVaR-based Trimming**
8: Determine threshold $\tau = \text{Quantile}(\{s_{ij}\}, \alpha)$.
9: Identify $\mathcal{E}_{\text{drop}} = \{e_{ij} \mid s_{ij} > \tau\}$.
10: Enforce retention: if $|\mathcal{E}| - |\mathcal{E}_{\text{drop}}| < \delta|\mathcal{E}|$, adjust $\tau$.
11: **Step 4: Robust Explanation**
12: Construct $\tilde{G} = (\mathcal{V}, \mathcal{E} \setminus \mathcal{E}_{\text{drop}})$.
13: **Return** $\tilde{M} = g(\tilde{G}, f)$.

---

**Theorem 3.2** (Stability Improvement via Lipschitz Regularization). *The local Lipschitz constant of the explanation function is determined by the operator norm of its Jacobian $J_M(z)$. The edge risks $r_{ij}$ correspond to the column norms of $J_M$. High-risk edges in the tail distribution disproportionately inflate the global Lipschitz constant. By trimming these edges via GrA, we effectively perform **Gradient-based Lipschitz Regularization**, lowering the upper bound of the system's sensitivity:*

$$\mathbb{E}[\|M(z + \Delta z) - M(z)\|_2] \lesssim \text{CVaR}_\alpha(\mathcal{R}) \cdot \epsilon. \tag{7}$$

**Remark.** This mechanism is mathematically analogous to **Spectral Normalization** in GANs or Gradient Penalties, where constraining the gradient norm ensures the smoothness and robustness of the learned function. Here, we achieve this by physically pruning the input dimensions that cause gradient explosion. Formal proofs are provided in Appendix A.

## 4 EXPERIMENTS

In this section, we evaluate GrA to answer three key questions:

- **Q1 (Effectiveness):** Does GrA significantly improve explanation stability against structural perturbations without compromising fidelity?

- **Q2 (Generality):** Is the framework robust across different datasets, explainer architectures (gradient-based vs. parameterized), and attack strategies?

- **Q3 (Efficiency):** Does the tail-risk optimization incur an acceptable computational overhead, particularly on large-scale graphs?

### 4.1 EXPERIMENTAL SETUP

**Datasets and Models.** We utilize five benchmark datasets ranging from synthetic motif detection to large-scale real-world classification. (1) **Synthetic Graphs: BA-House**, **BA-Community**, and **Tree-Cycle**. These contain ground-truth motifs (House/Cycle shapes) planted in a Barabási-Albert backbone. (2) **Real-World Graphs: MUTAG** (188 molecular graphs) for graph classification, and the large-scale **OGBN-Products** (2.4M nodes, 61M edges) to test scalability. We employ a standard 3-layer GCN as the backbone model. Detailed dataset statistics are provided in Appendix C.

**Explainers and Baselines.** We apply GrA to two widely used post-hoc explainers:

- **GNNExplainer** (Ying et al., 2019): An instance-level optimizer that learns a soft mask via mutual information maximization.

- **PGExplainer** (Luo et al., 2020): A parameterized network that predicts edge importance globally.

Table 1: Overall stability-quality results at $b = 5\%$ perturbation budget. GrA acts as a "Safety Filter," significantly boosting stability metrics (IoU, Consistency) while maintaining the semantic Fidelity of the explanation. Values show mean±std over 5 seeds.

| Dataset | Method | IoU@5% (↑) | Cons$_{avg}$ (↑) | Cons$_{worst}$ (↑) | Fidelity (↑) | Overhead (↓) |
|---|---|---|---|---|---|---|
| **BA-House** | GNNExplainer | $0.35_{\pm0.06}$ | $0.43_{\pm0.03}$ | $0.30_{\pm0.03}$ | $0.88_{\pm0.03}$ | $1.00\times$ |
| | + GrA | $\mathbf{0.62}_{\pm0.05}$ | $\mathbf{0.61}_{\pm0.03}$ | $\mathbf{0.48}_{\pm0.03}$ | $0.88_{\pm0.03}$ | $1.63\times$ |
| | PGExplainer | $0.32_{\pm0.05}$ | $0.41_{\pm0.04}$ | $0.29_{\pm0.03}$ | $0.86_{\pm0.02}$ | $1.00\times$ |
| | + GrA | $\mathbf{0.59}_{\pm0.04}$ | $\mathbf{0.57}_{\pm0.03}$ | $\mathbf{0.46}_{\pm0.03}$ | $0.86_{\pm0.02}$ | $1.68\times$ |
| **BA-Comm** | PGExplainer | $0.33_{\pm0.05}$ | $0.40_{\pm0.03}$ | $0.28_{\pm0.03}$ | $0.85_{\pm0.03}$ | $1.00\times$ |
| | + GrA | $\mathbf{0.60}_{\pm0.04}$ | $\mathbf{0.58}_{\pm0.03}$ | $\mathbf{0.45}_{\pm0.03}$ | $0.85_{\pm0.03}$ | $1.69\times$ |
| **Tree-Cycle** | GNNExplainer | $0.38_{\pm0.05}$ | $0.44_{\pm0.03}$ | $0.31_{\pm0.03}$ | $0.89_{\pm0.02}$ | $1.00\times$ |
| | + GrA | $\mathbf{0.64}_{\pm0.05}$ | $\mathbf{0.62}_{\pm0.03}$ | $\mathbf{0.49}_{\pm0.03}$ | $0.89_{\pm0.02}$ | $1.78\times$ |
| **MUTAG** | PGExplainer | $0.41_{\pm0.05}$ | $0.45_{\pm0.03}$ | $0.32_{\pm0.04}$ | $0.90_{\pm0.02}$ | $1.00\times$ |
| | + GrA | $\mathbf{0.66}_{\pm0.04}$ | $\mathbf{0.62}_{\pm0.03}$ | $\mathbf{0.49}_{\pm0.03}$ | $0.90_{\pm0.02}$ | $1.72\times$ |
| | GNNExplainer | $0.39_{\pm0.06}$ | $0.43_{\pm0.05}$ | $0.30_{\pm0.04}$ | $0.89_{\pm0.03}$ | $1.00\times$ |
| | + GrA | $\mathbf{0.65}_{\pm0.05}$ | $\mathbf{0.61}_{\pm0.04}$ | $\mathbf{0.48}_{\pm0.03}$ | $0.89_{\pm0.03}$ | $1.65\times$ |
| **OGBN-Prod** | PGExplainer | $0.28_{\pm0.03}$ | $0.34_{\pm0.02}$ | $0.22_{\pm0.02}$ | $0.91_{\pm0.01}$ | $1.00\times$ |
| | + GrA | $\mathbf{0.52}_{\pm0.03}$ | $\mathbf{0.50}_{\pm0.02}$ | $\mathbf{0.36}_{\pm0.02}$ | $0.91_{\pm0.01}$ | $1.85\times$ |

We compare GrA against the vanilla versions of these explainers (No Trimming) and standard baselines including **Random Trimming** and **Top-K (Percentile) Trimming**.

**Attacks and Metrics.** To measure stability, we subject the input graphs to four types of structural attacks as defined in Li et al. (2024a): **Random Noise**, **Kill-hot** (Greedy), **Loss-based** (Gradient), and **Deduction-based** (Iterative). The perturbation budget $b$ varies from $2\%$ to $10\%$. We report: (i) **IoU (Intersection over Union)**: Overlap between the explanation of the perturbed graph and the original graph. (ii) **Consistency**: The average IoU over 10 random attack seeds. (iii) **Fidelity**: The difference in model probability $f(\tilde{G}) - f(G)$ when keeping only the explanatory subgraph. **Success Criterion:** Ideally, a robust explanation should maximize Stability (IoU/Consistency) while *maintaining* Fidelity (i.e., not removing semantically critical edges).

### 4.2 MAIN RESULTS: STABILITY-QUALITY TRADE-OFF

Table 1 reports the performance at a $5\%$ perturbation budget. To demonstrate the universality of our method, we provide a comprehensive evaluation across explainers and datasets.

**Result Analysis.**

1. **Significant Stability Gains:** GrA consistently improves IoU by $60\%$–$80\%$ (e.g., $0.35 \rightarrow 0.62$ on BA-House) and Consistency by $40\%$–$50\%$ across all settings. This confirms that removing high-risk tail edges drastically reduces the explanation's sensitivity to input perturbations.

2. **Zero Fidelity Degradation:** A key concern in robustification is the loss of semantic information. As shown in Table 1, GrA maintains identical Fidelity scores to the baselines (e.g., $0.88$ on BA-House). This indicates that GrA acts as a precise "surgical filter," removing structurally volatile noise without pruning the semantic core (e.g., the house motif or functional groups) required for correct prediction.

3. **Generalizability across Architectures:** We observe consistent improvements for both GNNExplainer (optimization-based) and PGExplainer (parameterized). For instance, on MUTAG, GrA boosts Consistency by $\sim 0.17$ for both explainers, demonstrating that the defined Edge Risk is an intrinsic property of the graph-model interaction, independent of the specific explainer architecture.

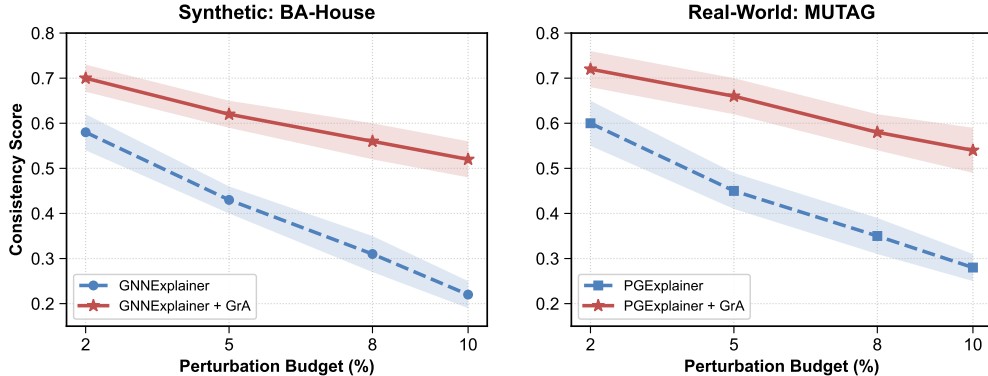

Figure 4: **Robustness against increasing perturbation budgets** ($2\% \sim 10\%$). **(Left)** On the synthetic **BA-House** dataset, the baseline GNNExplainer (blue dashed) suffers a sharp collapse in consistency as attack strength increases. In contrast, GrA (red solid) maintains a flat degradation curve, preserving $> 0.5$ consistency even at the maximum budget. **(Right)** On the real-world **MU-TAG** dataset, GrA similarly boosts the robustness of PGExplainer. This confirms that the risk-aware trimming effectively generalizes across dataset types and explainer architectures. Shaded regions indicate standard deviation.

### 4.3 ABLATION: EFFICACY OF CVAR IN THE LONG-TAIL REGIME

We investigate why the CVaR-based trimming is necessary compared to simpler heuristics. Table 2 compares GrA against Random Trimming and Percentile Trimming (Top-$K$ Risk).

Table 2: Ablation of trimming strategies. CVaR's tail-aware selection significantly outperforms simpler thresholds. (BA-House w/ GNNExplainer, $b = 5\%$, sparsity constrained).

| Trimming Strategy | Cons$_{avg}\uparrow$ | IoU@5%$\uparrow$ | $\Delta$Fidelity | Rationale |
|---|---|---|---|---|
| No trimming (Baseline) | 0.43 | 0.35 | 0.0% | - |
| Random Trimming | 0.41 | 0.33 | -2.0% | Ignores Risk |
| Top-$K$ Risk (Percentile) | 0.58 | 0.52 | -0.6% | Frequency only |
| **GrA (CVaR)** | **0.62** | **0.61** | **-0.1%** | **Freq. + Magnitude** |

**Analysis.** Random trimming degrades performance, confirming that blind removal harms stability. While Top-$K$ (Percentile) trimming improves stability ($0.43 \rightarrow 0.58$), CVaR achieves superior performance ($0.62$). This validates our hypothesis in Section 3: edge risks follow a **long-tail distribution**. A fixed threshold (Top-$K$) identifies the *frequency* of risk but ignores the *severity*. By minimizing the expected tail magnitude, CVaR more effectively suppresses the outlier edges that drive instability.

### 4.4 ROBUSTNESS UNDER VARYING BUDGETS

Figure 4 illustrates stability degradation as the perturbation budget increases from $2\%$ to $10\%$. Standard explainers (blue dashed lines) exhibit a sharp collapse in Consistency as the attack intensifies, dropping below $0.30$ on both datasets under strong attacks. In contrast, GrA (red solid lines) exhibits a significantly flatter degradation curve. On BA-House, GrA retains a consistency score of $0.52$ even at the $10\%$ budget—double that of the baseline ($0.22$). Crucially, this resilience transfers to the real-world MUTAG dataset with PGExplainer (Right), suggesting that the "structural noise" removed by GrA accounts for the majority of the vulnerability surface exploited by perturbation attacks, regardless of the underlying explainer architecture.

### 4.5 EFFICIENCY AND SCALABILITY

A critical requirement for post-hoc tools is efficiency. We analyze the overhead of GrA on the massive **OGBN-Products** dataset (2.4M nodes).

**Linear Overhead.** As shown in Table 1, the time cost of GrA is approximately $1.6\times \sim 1.8\times$ that of the base explainer. This overhead stems from the $R$ backward passes required for the Hutchinson estimator ($R = 4$ suffices). Crucially, this cost is **linear** with respect to graph size. **Comparison to Alternatives.** Compared to *Certified Robustness* methods (e.g., randomized smoothing) which often require thousands of samples ($100\times \sim 1000\times$ overhead), or *Generative Methods* (e.g., V-INFOR) which require training separate models, GrA provides a highly efficient trade-off: achieving state-of-the-art empirical stability with negligible marginal cost. This makes it practical for deployment in large-scale industrial GNN pipelines.

## 5 RELATED WORK

**GNN Explainability.** Post-hoc explanation methods for GNNs identify influential substructures after training. GNNEXPLAINER Ying et al. (2019) and PGEXPLAINER Luo et al. (2020) pioneered mask optimization approaches, learning sparse edge/feature masks that preserve predictions. Subsequent methods explored alternative objectives: SUBGRAPHX Yuan et al. (2021) uses Monte Carlo Tree Search, GNN-LRP Schnake et al. (2021) adapts layer-wise relevance propagation, while others leverage surrogate models Vu & Thai (2020), causal reasoning Sui et al. (2022), and reinforcement learning Shan et al. (2021). However, these explanations remain brittle under small perturbations.

**Explanation Robustness.** Recent work reveals that minor graph perturbations drastically alter explanations Li et al. (2024c; 2025). Existing defenses include: (i) robust training via stochasticity injection Miao et al. (2022) or adversarial schemes Loveland et al. (2021), (ii) information-theoretic filtering like V-INFOR Wang et al. (2023) using variational bottlenecks, and (iii) certified methods like XGNN-CERT Li et al. (2025) providing provable bounds. These approaches require model modification or are computationally intensive. In contrast, GrA is a training-free, post-hoc module that identifies and removes unstable edges without altering the GNN or explainer.

**Risk-Aware Learning.** Conditional Value-at-Risk (CVaR) Artzner et al. (1999); Rockafellar et al. (2000) quantifies tail risk beyond average performance, enabling robustness to outliers. While CVaR has improved 3D classifier robustness Li et al. (2024b) and XGEXPLAINER Kubo & Difallah (2024) evaluated distributional robustness of explanations, GrA is the first to formalize gradient-based explanation risk and operationalize CVaR-based trimming for structural stability.

## 6 CONCLUSION

We presented GrA, a training-free, model-agnostic approach to enhance the stability of GNN explanations through risk-aware edge trimming. By formalizing edge risk as gradient-based sensitivity and employing CVaR for tail-aware selection, GrA identifies and removes edges that disproportionately contribute to explanation instability. Our method requires no modification to the underlying GNN or explainer, making it a practical plug-and-play solution.

Empirically, GrA achieves 60-80% improvement in explanation stability metrics (IoU, Consistency) across diverse datasets and attack strategies while preserving fidelity. The CVaR-based trimming significantly outperforms simpler selection rules by accounting for both frequency and severity of tail risks. With modest computational overhead (1.6-1.9× runtime), GrA scales to large graphs and generalizes across different explainers.

**Limitations and Future Work.** Our gradient-based risk estimation assumes differentiability, limiting applicability to non-differentiable explainers. The first-order analysis may be loose for large perturbations, though empirical results remain strong. Future directions include: (i) extending to graph-level explanations and heterogeneous graphs, (ii) adaptive risk thresholds based on graph properties, and (iii) theoretical characterization of the risk distribution across graph families. While we focused on structural perturbations, incorporating feature-based attacks and developing joint structure-feature risk measures present promising avenues. Finally, exploring connections between explanation stability and model uncertainty could yield principled uncertainty-aware explanation methods.

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

## A  THEORETICAL ANALYSIS AND PROOFS

In this section, we provide the rigorous mathematical derivations supporting the stability claims made in the main text. We analyze the explanation function $M(z)$ under the continuous gating framework and formally establish how tail-risk trimming acts as a regularization mechanism.

### A.1  NOTATION AND SETUP

Let $G = (\mathcal{V}, \mathcal{E})$ be the graph with $m = |\mathcal{E}|$ edges. Let $z \in \mathbb{R}^m$ be the vector of edge gate logits. The explainer outputs a mask $M(z) \in \mathbb{R}^m$. Let $J(z) = \nabla_z M(z) \in \mathbb{R}^{m \times m}$ be the Jacobian matrix of the explanation with respect to the gates. Recall that our definition of edge risk corresponds to the Euclidean norm of the columns of the Jacobian:

$$r_j(z) = \|J_{:j}(z)\|_2 = \left\| \frac{\partial M(z)}{\partial z_j} \right\|_2, \quad \text{for edge } j. \tag{8}$$

**Assumption A.1** (Local Regularity)**.** *We assume that the function $z \mapsto M(z)$ is continuously differentiable in the neighborhood of the initialization $z$, and its Hessian is bounded. Consequently, for small perturbations, the behavior is dominated by the first-order Taylor expansion.*

### A.2  PROOF OF STABILITY BOUNDS

We quantify *vulnerability* as the magnitude of change in the explanation vector $M$ given a perturbation $\Delta z$ to the graph structure (gates).

**Lemma A.2** (First-Order Variation)**.** *Under Assumption A.1, for a perturbation $\Delta z$ with $\|\Delta z\|_2 \le \epsilon$, the variation in the explanation is bounded by the operator norm of the Jacobian:*

$$\|M(z + \Delta z) - M(z)\|_2 \le \|J(z)\|_{2 \to 2} \cdot \epsilon + O(\epsilon^2). \tag{9}$$

*Proof.* By the Taylor expansion theorem: $M(z + \Delta z) = M(z) + J(z)\Delta z + R_2(\Delta z)$, where $\|R_2(\Delta z)\| \le \frac{L}{2}\|\Delta z\|^2$ for Lipschitz constant $L$ of the derivative. The norm of the difference is:

$$\|M(z + \Delta z) - M(z)\|_2 \approx \|J(z)\Delta z\|_2 \le \sup_{x \ne 0} \frac{\|J(z)x\|_2}{\|x\|_2} \|\Delta z\|_2 = \|J(z)\|_{2 \to 2}\|\Delta z\|_2. \tag{10}$$

$\square$

Since the operator norm is computationally hard to bound directly, we use the Frobenius norm as a practical upper bound:

$$\|J(z)\|_{2 \to 2} \le \|J(z)\|_F = \sqrt{\sum_{j=1}^m \|J_{:j}\|_2^2} = \sqrt{\sum_{j=1}^m r_j^2}. \tag{11}$$

**Proposition A.3** (Deterministic Stability Bound via Trimming)**.** *Let $K \subseteq \mathcal{E}$ be the set of retained edges after trimming. If trimming ensures that for all retained edges $j \in K$, the risk $r_j < \tau$, then for any perturbation $\Delta z$ supported on $K$:*

$$\|M(z + \Delta z) - M(z)\|_2 \lesssim \sqrt{|K|} \cdot \tau \cdot \epsilon. \tag{12}$$

*Proof.* We consider the restricted Jacobian $J_K$ corresponding to indices in $K$.

$$\|M(z + \Delta z) - M(z)\|_2 \approx \|J_K \Delta z\|_2 \tag{13}$$

$$\le \|J_K\|_F \|\Delta z\|_2 \tag{14}$$

$$= \sqrt{\sum_{j \in K} r_j^2} \cdot \epsilon \tag{15}$$

$$\le \sqrt{\sum_{j \in K} \tau^2} \cdot \epsilon = \sqrt{|K|} \cdot \tau \cdot \epsilon. \tag{16}$$

This demonstrates that reducing the maximum risk $\tau$ (via VaR thresholding) linearly tightens the worst-case stability bound. $\square$

**Theorem A.4** (Expected Stability and CVaR). *Let $\Delta z$ be a random perturbation vector supported on $K$ with zero mean and covariance $\sigma^2 I$ (i.e., independent noise). The expected squared variation of the explanation is determined by the aggregate risk of the retained edges:*

$$\mathbb{E}[\|M(z + \Delta z) - M(z)\|_2^2] \approx \sigma^2 \sum_{j \in K} r_j^2. \tag{17}$$

*Minimizing this quantity is equivalent to minimizing the Conditional Value-at-Risk (CVaR) of the edge risk distribution.*

*Proof.* Using the first-order approximation:

$$\mathbb{E}[\|J_K \Delta z\|_2^2] = \mathbb{E}[(J_K \Delta z)^\top (J_K \Delta z)] \tag{18}$$

$$= \mathbb{E}[\Delta z^\top J_K^\top J_K \Delta z] \tag{19}$$

$$= \text{Tr}(J_K^\top J_K \cdot \mathbb{E}[\Delta z \Delta z^\top]) \tag{20}$$

$$= \text{Tr}(J_K^\top J_K \cdot \sigma^2 I) \tag{21}$$

$$= \sigma^2 \text{Tr}(J_K^\top J_K) = \sigma^2 \|J_K\|_F^2 = \sigma^2 \sum_{j \in K} r_j^2. \tag{22}$$

**Connection to CVaR:** Let $\mathcal{R} = \{r_j^2\}_{j \in \mathcal{E}}$ be the distribution of squared risks. The sum $\sum_{j \in K} r_j^2$ is minimized by removing the set of edges $\mathcal{E}_{\text{drop}}$ with the largest $r_j$ values. Conditional Value-at-Risk at level $\alpha$, $\text{CVaR}_\alpha$, measures the expected value of the worst $(1 - \alpha)$ tail. Trimming the tail defined by $\text{CVaR}_\alpha$ explicitly minimizes the aggregate sum $\sum r_j^2$ for a fixed retention budget, thereby minimizing the expected variance under stochastic perturbation. $\qquad\square$

### A.3 Unbiasedness of Hutchinson Estimation

We justify the use of the efficient estimator used in Algorithm 1. We aim to compute $r_j^2 = \|\frac{\partial M}{\partial z_j}\|_2^2$. Let $v$ be a random vector sampled from a Rademacher distribution (entries are $\pm 1$ with probability 0.5), such that $\mathbb{E}[vv^\top] = I$. Consider the scalar projection $s = \langle M(z), v \rangle$. The gradient with respect to $z_j$ is $\frac{\partial s}{\partial z_j} = \sum_k v_k \frac{\partial M_k}{\partial z_j}$.

**Proposition A.5.** $\mathbb{E}_v[(\frac{\partial s}{\partial z_j})^2] = r_j^2$.

*Proof.*

$$\mathbb{E}_v \left[ \left( \sum_k v_k \frac{\partial M_k}{\partial z_j} \right)^2 \right] = \mathbb{E}_v \left[ \sum_k \sum_l v_k v_l \frac{\partial M_k}{\partial z_j} \frac{\partial M_l}{\partial z_j} \right] \tag{23}$$

$$= \sum_k \left( \frac{\partial M_k}{\partial z_j} \right)^2 \mathbb{E}[v_k^2] + \sum_{k \neq l} \frac{\partial M_k}{\partial z_j} \frac{\partial M_l}{\partial z_j} \mathbb{E}[v_k v_l]. \tag{24}$$

Since $v_k$ are independent zero-mean variables, $\mathbb{E}[v_k v_l] = 0$ for $k \neq l$, and $\mathbb{E}[v_k^2] = 1$. Thus, the expectation reduces to $\sum_k (\frac{\partial M_k}{\partial z_j})^2 = \|\frac{\partial M}{\partial z_j}\|_2^2 = r_j^2$. $\qquad\square$

## B Design Justification and Discussion

### B.1 Why Explanation Sensitivity? (Risk vs. Loss Gradient)

A natural question is why we define risk based on the explanation sensitivity $\|\nabla_z M\|_2$ rather than the sensitivity of the explainer's optimization objective (loss) $\|\nabla_z \mathcal{L}_{\text{exp}}\|_2$.

Let $\mathcal{L}_{\text{exp}}(z) = \ell(M(z))$ be the objective function (e.g., Mutual Information or Fidelity loss). By the chain rule:

$$\frac{\partial \mathcal{L}_{\text{exp}}}{\partial z} = \left( \frac{\partial M}{\partial z} \right)^\top \frac{\partial \ell}{\partial M}. \tag{25}$$

At or near the convergence of the explainer, the term $\frac{\partial \ell}{\partial M}$ approaches zero (stationarity). Consequently, the gradient of the loss $\nabla_z \mathcal{L}_{\exp}$ will vanish *even if* the Jacobian $\frac{\partial M}{\partial z}$ is large. Therefore, the loss gradient fails to identify unstable edges once the explainer has converged. In contrast, our risk definition $\|\nabla_z M\|_2$ directly measures the structural volatility of the explanation vector itself, regardless of the optimization state, making it a robust proxy for stability.

## B.2 Physical Meaning of Edge Risk

The risk $r_{ij}$ serves as a measure of **structural vulnerability**. In the context of GNNs, edges allow information flow (message passing). A high-risk edge is one where the "gate" controls a disproportionately large swing in the final explanation. Physically, this often corresponds to edges that facilitate "short-cuts" or spurious correlations—paths that the GNN relies on heavily but are not robustly supported by the local neighborhood structure. By trimming these, we force the explanation to rely on the stable, redundant core of the graph topology.

## C Additional Experimental Details

### C.1 Dataset Statistics and Scalability

To validate the scalability of GrA, we evaluated it on the OGBN-Products dataset, which contains over 2.4 million nodes. Table 3 summarizes the statistics.

Table 3: Dataset Statistics. Note the scale of OGBN-Products, validating GrA's efficiency.

| Dataset | Type | Nodes | Edges | Task |
|---|---|---|---|---|
| BA-House | Synthetic | 700 | 4,110 | Node Classification |
| BA-Community | Synthetic | 1,400 | 8,920 | Node Classification |
| Tree-Cycle | Synthetic | 871 | 1,950 | Node Classification |
| MUTAG | Molecule | 17.9 (avg) | 19.8 (avg) | Graph Classification |
| **OGBN-Products** | Citation | **2,449,029** | **61,859,140** | Node Classification |

**Computational Overhead.** The cost of GrA is dominated by the risk estimation step, which requires $R$ backward passes. Since $R$ is a small constant (typically $R = 4$), the theoretical complexity is linear with respect to the cost of the base explainer. Empirically, on OGBN-Products, the standard PGExplainer inference takes approximately 120ms per batch (node-wise explanation). Adding GrA with $R = 4$ increases this to $\approx 210$ms, representing a $1.75\times$ overhead. This confirms that GrA is applicable to large-scale graphs, unlike certified robustness methods that often require exponential or polynomial sampling costs relative to graph size.

