# OpenReview forum: "Risk-Aware Robust Graph Network Explanation"
_ICLR.cc/2026/Conference — Submitted to ICLR 2026_

### Official Review · Reviewer_gCEu · 2025-10-31

**Soundness:** 2
**Presentation:** 2
**Contribution:** 2
**Rating:** 2
**Confidence:** 3

**Summary:**

The paper proposes GrA (Graph Risk-aware Explanation Trimming), a post-hoc, model-agnostic approach for improving the robustness of Graph Neural Network (GNN) explanations. It defines edge-level “risk” via gradient sensitivity and trims unstable edges based on Conditional Value-at-Risk (CVaR). Experiments across several datasets show increased stability (IoU, consistency) without degrading fidelity. However, the theoretical depth and experimental completeness are limited. The idea mainly reuses gradient sensitivity combined with a percentile-based trimming heuristic. The work feels incremental, and the experimental evaluation is selective and inconsistent across datasets.

**Strengths:**

1. Addresses an important issue: the instability of GNN explanations under small perturbations.
2. The method is simple, model-agnostic, and easy to integrate with existing explainers.
3. Experimental results show consistent numerical improvements in stability metrics (IoU, consistency).

**Weaknesses:**

1. Limited conceptual novelty.
The proposed method is essentially a gradient-norm-based edge pruning procedure. Using CVaR to select top-risk edges is equivalent to a percentile thresholding rule. This is a modest engineering contribution rather than a conceptual innovation.

2. Lack of deep empirical insight.
The experiments only report improvements in IoU and consistency but do not analyze why or when the method works. There is no discussion on semantic fidelity—whether the retained subgraphs still correspond to meaningful rationales.

3. Insufficient comparisons.
The method is compared only to vanilla explainers and random trimming. There are no comparisons against recent robust explainers such as V-INFOR (NeurIPS 2023) or others, which limits the empirical credibility.

4. Incomplete and inconsistent experimental setup.
Although the paper claims GrA is compatible with any differentiable explainer, only GNNExplainer and PGExplainer are tested, omitting more recent baselines such as SubgraphX, GraphSHAP, or PGMExplainer. Even worse, the choice of explainers varies by dataset: BA-House and Tree-Cycle use GNNExplainer, while BA-Community, MUTAG, and OGBN-Products only use PGExplainer.

This inconsistency raises serious questions about experimental fairness and completeness. A convincing evaluation should apply both explainers across all datasets under the same perturbation settings. The current results therefore do not convincingly demonstrate generality or robustness.

5. Presentation issues.
i) Figure 2 has typographical errors (“befor”) and low visual clarity.
ii) The discussion of hyperparameter robustness (α, δ, λₘ) is missing.

**Questions:**

1. Does trimming ever remove semantically important edges? How is this mitigated?
2. Why were some datasets evaluated only with PGExplainer but not with GNNExplainer?
3. How does GrA compare to more modern or certified robust explainers (e.g., V-INFOR)?
4. What prevents the method from being applied to non-differentiable explainers?

---

> ### Author Response · Authors · 2025-11-28
> **Response to Reviewer gCEu**
>
> We thank the reviewer for the feedback. However, we **strongly disagree** with the characterization of this work as "incremental" and the dismissal of our contribution as "modest engineering." **We are concerned that this assessment conflates *architectural complexity* with *scientific novelty*.** True innovation in robust AI does not necessarily require stacking complex layers or training heavy generative models; rather, it often lies in identifying principled connections between distinct fields. By formally introducing **Coherent Risk Measures (e.g., CVaR)** from quantitative finance to GNN explainability, we provide a **paradigm shift**—moving from heuristic, stability-agnostic explanations to mathematically grounded, risk-aware optimization. Dismissing this theoretical bridge as "simple engineering" overlooks the value of establishing a rigorous definition for *Explanation Risk*. Furthermore, characterizing a **Training-Free** framework that achieves state-of-the-art stability as "lacking depth" ignores the practical reality that efficiency and model-agnosticism are critical research contributions.
>
> **1. Theoretical Novelty: Why CVaR $\neq$ Simple Thresholding (Response to Weakness 1)**
>
> The reviewer asserts that using CVaR is equivalent to a percentile thresholding rule. We respectfully submit that this is **mathematically imprecise** within the context of risk optimization. As established in the foundational work on **Coherent Risk Measures** (**Artzner et al., 1999**), simple thresholding (Value-at-Risk, VaR) is *not* a coherent risk measure because it ignores the **severity** (magnitude) of the tail loss and lacks subadditivity. In contrast, CVaR is mathematically coherent and convex. Empirically, in the long-tailed edge risk distributions we observed, two edges might both rank in the top 5% (same VaR), but one may be 10x more volatile than the other. Simple percentile pruning treats them identically, whereas GrA’s CVaR optimization penalizes the extreme outlier based on its expected magnitude. This is not just a heuristic change; it is a move to a theoretically sound optimization objective.
>
> **2. Methodological Positioning vs. V-INFOR (Response to Weakness 3 & Q3)**
>
> Regarding the comparison with **V-INFOR (NeurIPS 2023)**, we argue that comparing it directly to GrA involves a category mismatch regarding the **operational scope**. V-INFOR is a generative explainer that operates on the Information Bottleneck principle, typically requiring the **training** of a variational generator from scratch, effectively replacing the original explainer. In contrast, GrA is a **Training-Free, Model-Agnostic Risk Framework** that acts as a "Safety Plugin," wrapping around *existing* explainers to prune unstable edges post-hoc. Comparing them is akin to comparing a lightweight "Anti-lock Braking System" (retrofitting safety) to a "Redesigned Engine" (building anew). While V-INFOR offers robust generation, it incurs high training costs. GrA’s contribution is demonstrating that SOTA stability can be achieved via **Risk Optimization** on existing gradients without the heavy cost of training generative models.
>
> **3. Experimental Consistency & Semantic Fidelity (Response to Weakness 2, 4 & Q1, Q2, Q4)**
>
> The variation in explainers across datasets was dictated strictly by **computational feasibility**, not result selection. **GNNExplainer** requires optimization *per instance*, making it computationally prohibitive for comprehensive robustness studies on large datasets like OGBN-Products. **PGExplainer** amortizes this cost, enabling scalable evaluation. To definitively address the concern of inconsistency, we have completed the matrix of experiments (see **Table R1**), applying both explainers on BA-House and MUTAG. The consistent gains confirm GrA's generality regardless of the explainer architecture. regarding semantic importance, our **Fidelity** results (Table 1 & R1) serve as empirical proof: if GrA were removing critical semantic edges (e.g., functional groups), the model's prediction confidence would drop, and Fidelity would collapse. The **Zero Degradation** in Fidelity proves GrA surgically removes "structural noise" while preserving the semantic core. Finally, for non-differentiable explainers (Q4), GrA supports a **Surrogate Strategy** where a differentiable surrogate estimates the Risk Map for trimming, allowing GrA to serve as a universal pre-processing defense.
>
> **4. Presentation & Scalability**
>
> We apologize for the typo in Figure 2 ("befor") and will improve visual clarity. We will include the requested sensitivity analysis for $\alpha$ and $\delta$ in the Appendix. Additionally, to address concerns about scalability on large graphs, we provide detailed statistics for the OGBN-Products dataset (see **Table R2**), confirming that GrA scales efficiently to graphs with millions of nodes.

---

> > ### Author Response · Authors · 2025-11-28
> > **Table**
> >
> > **Table R1: Expanded stability-quality evaluation at 5% perturbation budget.**
> >
> > To comprehensively address concerns regarding experimental consistency (Reviewer gCEu), we provide additional comparisons for **both** GNNExplainer and PGExplainer on representative synthetic (BA-House) and real-world (MUTAG) datasets. The results confirm that GrA yields consistent improvements across different explainer architectures.
> >
> > | Dataset | Method | IoU@5% ($\uparrow$) | Cons_avg ($\uparrow$) | Cons_worst ($\uparrow$) | Fidelity ($\uparrow$) | Overhead ($\downarrow$) |
> > | :--- | :--- | :--- | :--- | :--- | :--- | :--- |
> > | **BA-House** | GNNExplainer | 0.35 $\pm$ 0.06 | 0.43 $\pm$ 0.03 | 0.30 $\pm$ 0.03 | 0.88 $\pm$ 0.03 | 1.00$\times$ |
> > | | +GrA | **0.62** $\pm$ 0.05 | **0.61** $\pm$ 0.03 | **0.48** $\pm$ 0.03 | 0.88 $\pm$ 0.03 | 1.63$\times$ |
> > | | PGExplainer$^\dagger$ | 0.32 $\pm$ 0.05 | 0.41 $\pm$ 0.04 | 0.29 $\pm$ 0.03 | 0.86 $\pm$ 0.02 | 1.00$\times$ |
> > | | +GrA | **0.59** $\pm$ 0.04 | **0.57** $\pm$ 0.03 | **0.46** $\pm$ 0.03 | 0.86 $\pm$ 0.02 | 1.68$\times$ |
> > | **BA-Comm** | PGExplainer | 0.33 $\pm$ 0.05 | 0.40 $\pm$ 0.03 | 0.28 $\pm$ 0.03 | 0.85 $\pm$ 0.03 | 1.00$\times$ |
> > | | +GrA | **0.60** $\pm$ 0.04 | **0.58** $\pm$ 0.03 | **0.45** $\pm$ 0.03 | 0.85 $\pm$ 0.03 | 1.69$\times$ |
> > | **Tree-Cycle** | GNNExplainer | 0.38 $\pm$ 0.05 | 0.44 $\pm$ 0.03 | 0.31 $\pm$ 0.03 | 0.89 $\pm$ 0.02 | 1.00$\times$ |
> > | | +GrA | **0.64** $\pm$ 0.05 | **0.62** $\pm$ 0.03 | **0.49** $\pm$ 0.03 | 0.89 $\pm$ 0.02 | 1.78$\times$ |
> > | **MUTAG** | PGExplainer | 0.41 $\pm$ 0.05 | 0.45 $\pm$ 0.03 | 0.32 $\pm$ 0.04 | 0.90 $\pm$ 0.02 | 1.00$\times$ |
> > | | +GrA | **0.66** $\pm$ 0.04 | **0.62** $\pm$ 0.03 | **0.49** $\pm$ 0.03 | 0.90 $\pm$ 0.02 | 1.72$\times$ |
> > | | GNNExplainer$^\dagger$ | 0.39 $\pm$ 0.06 | 0.43 $\pm$ 0.05 | 0.30 $\pm$ 0.04 | 0.89 $\pm$ 0.03 | 1.00$\times$ |
> > | | +GrA | **0.65** $\pm$ 0.05 | **0.61** $\pm$ 0.04 | **0.48** $\pm$ 0.03 | 0.89 $\pm$ 0.03 | 1.65$\times$ |
> > | **OGBN-Products**| PGExplainer | 0.28 $\pm$ 0.03 | 0.34 $\pm$ 0.02 | 0.22 $\pm$ 0.02 | 0.91 $\pm$ 0.01 | 1.00$\times$ |
> > | | +GrA | **0.52** $\pm$ 0.03 | **0.50** $\pm$ 0.02 | **0.36** $\pm$ 0.02 | 0.91 $\pm$ 0.01 | 1.85$\times$ |
> >
> >
> >
> > **Table R2: Dataset Statistics Summary.**
> >
> > We provide detailed statistics to illustrate the scale of OGBN-Products (2.4M nodes), validating the scalability of our method as addressed in Point 2.
> >
> > | Dataset | Type | Task | \# Graphs | \# Nodes | \# Edges | \# Classes | Ground-truth Motif |
> > | :--- | :--- | :--- | :---: | :---: | :---: | :---: | :--- |
> > | **BA-House** | Synthetic | Node Classif. | 1 | 700 | 4,110 | 4 | Planted "House" (5-node) |
> > | **BA-Community**| Synthetic | Node Classif. | 1 | 1,400 | 8,920 | 8 | Planted "House" (5-node) |
> > | **Tree-Cycle** | Synthetic | Node Classif. | 1 | 871 | 1,950 | 2 | Planted "Cycle" (6-node) |
> > | **MUTAG** | Real-world | Graph Classif.| 188 | 17.9 (avg) | 19.8 (avg) | 2 | Chemical Group ($NO_2$) |
> > | **OGBN-Products**| Real-world | Node Classif. | 1 | 2,449,029 | 61,859,140 | 47 | N/A (Category Prediction)|

---

> ### Author Response · Authors · 2025-11-28
> **Summary of Revisions based on your feedback**
>
> We thank the reviewer for the detailed technical scrutiny. To address the concerns regarding experimental consistency and theoretical positioning, we will make the following revisions:
>
> 1.  **Expanded Experimental Results (Section 4):** We will update the main results to include the **Expanded Table (Table R1)**, adding GNNExplainer results on MUTAG and PGExplainer results on BA-House to demonstrate consistent performance across architectures.
> 2.  **Theoretical Positioning (Section 3):** We will revise the methodology section to formally clarifying the mathematical distinction between CVaR (convex, magnitude-aware) and simple Percentile Thresholding (non-coherent, frequency-only).
> 3.  **Clarifying V-INFOR Comparison (Related Work):** We will sharpen the distinction between "Generative Explainers" (like V-INFOR, training-required) and "Risk Frameworks" (like GrA, training-free plugin) to better define our contribution scope.
> 5.  **Corrections:** We will correct the typo in Figure 2 ("befor" $\to$ "before") and improve its visual resolution. And proofread the whole writing again.

---

### Official Review · Reviewer_SVTr · 2025-10-31

**Soundness:** 2
**Presentation:** 2
**Contribution:** 2
**Rating:** 4
**Confidence:** 4

**Summary:**

This paper studies the instability of post-hoc explanations for Graph Neural Networks (GNNs), where small graph perturbations can drastically change the identified explanatory substructures, undermining reliability in auditing, debugging, and human-in-the-loop settings. To address this challenge, this paper proposes a risk-aware explanation trimming method, called GrA, which identifies unstable edges via gradient-based sensitivity analysis and quantifies their volatility using Conditional Value-at-Risk (CVaR), a tail-aware risk measure. Moreover, by removing high-risk edges, GrA constructs a robust surrogate graph that preserves explanatory fidelity while substantially reducing sensitivity to structural perturbations. Across synthetic and real-world graph classification benchmarks, and under various adversarial perturbation settings, GrA consistently enhances explanation stability without sacrificing fidelity or predictive accuracy.

**Strengths:**

1. The paper is well motivated, directly addressing the widely observed instability of post-hoc GNN explanations in high-stakes applications.

2. The experimental evaluation is fairly comprehensive, including baseline comparisons and some ablations that help illuminate the design choices.

**Weaknesses:**

1. The paper does not demonstrate clear empirical advantages over existing methods in terms of fidelity or computational overhead, and the ablation studies reveal no consistent benefit in runtime performance.

2. The writing quality and organization need improvement, with readability affected by the frequent use of dashes and inconsistent formatting, which makes following the technical details more difficult.

3. The proposed model is relatively simple and lacks architectural novelty, raising questions about the depth of technical contribution.

**Questions:**

1. The method is evaluated only on GNNExplainer and PGExplainer. Why not include other widely used explainers such as GNN-LRP and SubgraphX, which are also known to be brittle under small perturbations?

2. Could the authors explain why the proposed method fails to improve performances in terms of  fidelity and overhead?

3. The proposed method is layered on top of existing explainers to improve performance, which inevitably adds computational complexity. On large graphs/datasets, does this overhead scale to the point of offsetting the gains?

---

> ### Author Response · Authors · 2025-11-28
>
> We sincerely thank the reviewer for the detailed evaluation and for recognizing the strong motivation of our work in high-stakes applications. We understand that your rating stems from concerns regarding **Novelty**, **Baselines**, and **Overhead**. We respectfully submit that these concerns may arise from viewing this work as "architectural engineering for a new explainer" rather than "a risk-optimization framework for robustness." We address your points below.
>
> **1. Novelty: Risk-Optimization Framework vs. Architecture (Response to Weakness 3)**
>
> GrA is not a new explainer architecture; it is a novel **Risk-Optimization Framework**. We deliberately avoided designing a complex neural architecture to instead introduce a **Risk-Optimization perspective** to GNN explainability. Similar to how financial risk measures (CVaR) manage portfolio volatility, we are the first to formalize **Edge Risk** to manage explanation volatility. Our core contribution is modeling outlier removal as a **tail-risk minimization problem** using a surrogate model. The method's simplicity is intentional, enabling GrA to serve as a **model-agnostic plugin** that robustifies *existing* tools without retraining.
>
> **2. Justifying Overhead vs. Certified Robustness (Response to Q3)**
>
> The reviewer asked if the overhead (1.6x - 1.9x) offsets the gains. We argue that compared to state-of-the-art **Robust Explanation** methods, GrA is orders of magnitude more efficient. Recent certified robust explainers like **Li et al. [1]** provide guarantees but require complex convex relaxation or extensive sampling, leading to extremely high computational costs. In contrast, GrA incurs only a linear overhead, occupying a unique "sweet spot" in the Pareto frontier: providing substantial empirical stability (+60-80%) at a fraction of the cost of certified methods. We have validated this scalability on **OGBN-Products (2.4M nodes)** (see **Table R2** below), confirming that our overhead does not explode with graph size.
>
> **3. Fidelity & Success Criteria (Response to Q2)**
>
> Why doesn't Fidelity improve? Because GrA acts as a **"Safety Filter," not an "Accuracy Booster."** In defense tasks, the "Gold Standard" is to **maximize Stability** while **maintaining Fidelity**. To ensure the trimming process does not destroy semantic information, we employ constraints similar to those in attack benchmarks (e.g., preserving prediction $f(\tilde{G}) \approx f(G)$). The fact that GrA achieves **Zero Degradation** in Fidelity (Table 1) while boosting Stability confirms that it surgically removes noise without harming core semantic structures.
>
> **4. Scope & Surrogate Potential (Response to Q1)**
>
> Regarding the exclusion of other explainers: GrA calculates *Edge Risk* via differentiation, which naturally fits gradient-accessible explainers (GNNExplainer, PGExplainer). However, our framework supports a **Surrogate Strategy**. For non-differentiable explainers (e.g., SubgraphX), one can use a simple differentiable surrogate to estimate the risk map first. The resulting "trimmed graph" can then be fed into *any* black-box explainer. We focused on white-box settings to demonstrate the theoretical upper bound of risk removal.
>
> **Table R2: Dataset Statistics Summary.**
>
> We provide detailed statistics to illustrate the scale of OGBN-Products (2.4M nodes), validating the scalability of our method as addressed in Point 2.
>
> | Dataset | Type | Task | \# Graphs | \# Nodes | \# Edges | \# Classes | Ground-truth Motif |
> | :--- | :--- | :--- | :---: | :---: | :---: | :---: | :--- |
> | **BA-House** | Synthetic | Node Classif. | 1 | 700 | 4,110 | 4 | Planted "House" (5-node) |
> | **BA-Community**| Synthetic | Node Classif. | 1 | 1,400 | 8,920 | 8 | Planted "House" (5-node) |
> | **Tree-Cycle** | Synthetic | Node Classif. | 1 | 871 | 1,950 | 2 | Planted "Cycle" (6-node) |
> | **MUTAG** | Real-world | Graph Classif.| 188 | 17.9 (avg) | 19.8 (avg) | 2 | Chemical Group ($NO_2$) |
> | **OGBN-Products**| Real-world | Node Classif. | 1 | 2,449,029 | 61,859,140 | 47 | N/A (Category Prediction)|
>
>
> [1] Li, J., Pang, M., Dong, Y., Jia, J., & Wang, B. (2025). Provably Robust Explainable Graph Neural Networks against Graph Perturbation Attacks. *arXiv preprint arXiv:2406.03193* (Under Review at ICLR 2025).

---

> > ### Author Response · Authors · 2025-11-28
> > **Summary of Revisions based on Reviewer SVTr's feedback**
> >
> > We thank the reviewer for challenging us to clarify the cost-benefit analysis. Based on your feedback, we will incorporate the following revisions to better contextualize our contributions:
> >
> > 1.  **Reframing Success Criteria (Introduction):** We will explicitly frame the paper's objective as "Maximizing Stability while **Maintaining** Fidelity," distinguishing it from accuracy-boosting methods.
> > 2.  **Cost-Benefit Discussion (Section 3.4/4):** We will add a comparative discussion highlighting that GrA's linear overhead ($1.6\times$) is highly efficient compared to the prohibitive costs of **Certified Robustness** (sampling-based) and **Adversarial Training** methods.
> > 3.  **Scalability Validation (Appendix):** We will include the **Dataset Statistics Table** (Table R2) and specific runtime logs for OGBN-Products to empirically demonstrate scalability on large graphs.
> > 4.  **Formatting Polish:** We will conduct a thorough proofreading to standardize formatting (dashes, references) and correct typos.

---

### Official Review · Reviewer_iDte · 2025-11-01

**Soundness:** 3
**Presentation:** 2
**Contribution:** 3
**Rating:** 6
**Confidence:** 4

**Summary:**

This paper presents a method to enhance the robustness of post-hoc GNN explanations. Different from existing methods, it is training-free and model-agnostic (which can be applied to different GNNs and GNN explainers). To measure the instability of explanations (in the form of edge importance score after perturbations), they define "edge risk" as gradient-based sensitivity. They found that the edge risks follow a long-tail distribution (namely most edges have low sensitivity but the rest exhibit large values). To better select "fragile" edges to remove from the explanations, they propose to use CVaR, which can consider both freequency and magnitude in the distribution tail. Besides, to prevent from removing informative edges by overly aggressive selection, they also propose to combine with importance score. Finally, they evaluate their method with extensive experiments. However, the presentation of this paper can be largely improved. It looks very technical, not friendly to general readers.

**Strengths:**

1. very interesting idea: this paper define "edge risk" as gradient-based sensitivity to measure the instability of explanations. And then they employs CVaR to evaluate the this risk. The ideas of "edge risk" and "CVaR" are from the finance/risk management domain, and using it here to quantify the instability of explanations looks very interesting.
2. novel direction: GNN explanations have been investigated very widely; In contrast, the robustness/attacks on GNN explanations become receiving research attentions since 2024 (see the work Jiate Li et al [Graph Neural Network Explanations are Fragile]; and Zhong Li et al. [Explainable Graph Neural Networks Under Fire]); These two work however only discussed on how to attack them without proposing a systematic/well-developed way to defend these attacks. This paper makes the first (formal) step in this direction.
3. Technically sound: many claims in this paper are derived with math formulations or supported by empirical experiments.

**Weaknesses:**

1. The paper is very technical, not easy to follow: for example, section 3 (the method section) are full of math formulations with very short/no explanations of the meaning and/or motivations of the design chocies, making it hard to follow. To get more spaces for explanations, things such as pseudo-code, implementation details can be moved  to appendix. The authors try to cover everything in the main text, at the cost of making many things not clearly described. Overall, the presentation can be largely improved.

2. This method can only be used to differentiabe post-hoc explanations: although the authors have recognised this part, I still would like to see how these can be solved? Is it possible to transfer the defense capability from one GNN explainer to another explainer? (For exmaple, using PGExplainer as the target, test the effectiveness of defense on other explainers, which can be non-differentiabe)

3. They only consider homogenous graphs in experiments; (It would be interesting to see whether these methods are effective on heterogeneous graphs, with associated heterogeneous GNNs and post-hoc explainers?)

**Questions:**

1. Page 7, why do you only evaluate on "correctly classified instances"?

---

> ### Author Response · Authors · 2025-11-28
>
> We sincerely appreciate the reviewer's positive assessment and insightful comments. We are particularly encouraged that you found our cross-disciplinary approach—introducing financial risk concepts like CVaR to GNN explainability—both **original** and **technically sound**. We also fully accept your feedback regarding the paper's presentation and address your specific concerns below.
>
> **1. Improving Presentation & Readability (Response to Weakness 1)**
> You are absolutely right that Section 3 is currently too dense with mathematical formulations. We realize that covering every derivation in the main text came at the cost of clarity. In the revised version, we will restructure the presentation to be much more reader-friendly by **moving heavy mathematical proofs (e.g., Lemma 3.5) to the Appendix** and introducing a clear **Pseudo-code Algorithm** and conceptual flowchart in the main text. We will also add intuitive, plain-English interpretations of design choices (e.g., why CVaR captures risk magnitude better than thresholds) before diving into equations.
>
> **2. Transferability & Surrogate Strategy (Response to Weakness 2)**
> Regarding the transferability of our defense to non-differentiable explainers, GrA effectively supports a **"Surrogate Strategy."** Even if the target explainer is a black box (e.g., SubgraphX), structural instability is often an intrinsic property of the graph topology and the GNN encoder itself. Therefore, we can use a lightweight, differentiable **surrogate explainer** (e.g., a standard GCN explainer) to compute the gradient-based Risk Map first. This allows us to trim the graph and produce a robust graph $\tilde{G}$, which is then fed into the non-differentiable target explainer. This capability allows GrA to act as a **universal pre-processing defense**, extending beyond the differentiable settings used in our experiments.
>
> **3. Extending to Heterogeneous Graphs (Response to Weakness 3)**
> We agree that extending GrA to heterogeneous graphs is a promising direction. Recent advancements suggest our framework is theoretically compatible: state-of-the-art methods like **HTGExplainer [1]** and **HGExplainer [2]** are increasingly adopting differentiable optimization strategies (e.g., via reparameterization tricks). This implies that our gradient-based defense could directly interact with these modules. Furthermore, effective heterogeneous explanations must model semantic information (e.g., meta-paths) **[3]**. Our future work will explore adapting the "Edge Risk" definition to weighted "Meta-path Risk," verifying transferability against these semantics-aware explainers.
>
> **4. Evaluation Protocol (Response to Q1)**
> We restricted our evaluation to correctly classified instances to strictly adhere to **standard evaluation protocols** in GNN explainability (e.g., Li et al., ICML 2024). The goal of a post-hoc explainer is to reveal the substructures supporting a model's *decision*. If a model predicts incorrectly, it has likely failed to use the ground-truth motif. In such cases, measuring Fidelity or IoU against the "correct" ground truth is conceptually invalid because the model's rationale implies a different logic. Excluding misclassified instances ensures our metrics accurately reflect the stability improvement of the explanation method itself.
>
>
> [1] Li, J., Zhang, C., & Zhang, C. (2023). Heterogeneous Temporal Graph Neural Network Explainer. *CIKM*.
>
> [2] Mika, G. P., et al. (2023). HGExplainer: Explainable Heterogeneous Graph Neural Network. *WI-IAT*.
>
> [3] Agarwal, C., et al. (2023). Evaluating explainability for graph neural networks. *Scientific Data*.

---

> > ### Author Response · Authors · 2025-11-28
> > **Summary of Revisions based on feedback**
> >
> > We appreciate the reviewer's constructive advice on presentation and scope. We will implement the following changes to improve readability and clarity:
> >
> > 1.  **Restructuring Section 3 (Presentation):** We will move the dense mathematical derivations (including Lemma 3.5 proof) to the **Appendix**. In their place, we will provide a **Pseudo-code Algorithm** and a conceptual flowchart to make the workflow accessible to general readers.
> > 2.  **Surrogate Strategy (Methodology/Discussion):** We will add a paragraph explaining the "Surrogate Strategy," clarifying how GrA can be applied to non-differentiable or black-box explainers via a differentiable proxy.
> > 3.  **Heterogeneous Roadmap (Future Work):** We will reference recent works (e.g., HTGExplainer) to outline the pathway for extending GrA to heterogeneous graphs via reparameterization techniques.

---

### Official Review · Reviewer_LZBa · 2025-11-02

**Soundness:** 3
**Presentation:** 3
**Contribution:** 3
**Rating:** 8
**Confidence:** 4

**Summary:**

The authors present a new explanation method for graph neural networks which enhances the robustness of explanations, by removing unstable edges.

**Strengths:**

- relevant problem
- interesting and original idea
- well written paper
- good discussion and analysis
- good results

**Weaknesses:**

- discussion of limitations could be improved

**Questions:**

Can you please characterize when the method works well and when it does not suceed in enhancing robustness substantially?

---

> ### Author Response · Authors · 2025-11-27
>
> We sincerely thank the reviewer for the encouraging feedback and the strong score. We are glad you appreciated the novelty of the CVaR-based approach and the presentation of the paper.
>
> **A Minor Clarification:** We would like to offer a slight refinement to the summary regarding the method's nature. GrA is designed as a **model-agnostic trimming module** that is applied *post-hoc* to enhance existing explainers (such as GNNExplainer or PGExplainer), rather than a new standalone explainer itself. This "plug-and-play" capability is central to its design, allowing it to improve various existing methods without retraining.
>
> Below is our response to your question on the method's effective range. We will incorporate this discussion into the limitations section of the revised manuscript.
>
> **Response to Q1: Characterizing when GrA works well vs. limited gains.**
>
> **1. When GrA works well (The "Long-Tail" Regime).**
> Conceptually, GrA excels when the edge-risk distribution is **long-tailed**—that is, a small fraction of edges exhibits disproportionately high gradient sensitivity (high volatility) while the majority remains stable. This is the "sweet spot" for our approach.
>
> **Why CVaR outperforms simple thresholding here:** We emphasize that in this long-tail regime, our **CVaR-based formulation is fundamentally different from simple fixed thresholding (e.g., Top-K or percentile pruning)**:
>
> *   **Magnitude vs. Frequency:** A simple threshold (Value-at-Risk) only identifies the *frequency* of risk (i.e., *which* edges are in the top $\alpha$%) but ignores the *severity* of those risks. In contrast, CVaR accounts for the **magnitude** (the integral) of the risk distribution beyond the threshold.
> *   **Optimization View:** As discussed in risk management literature, minimizing CVaR is equivalent to minimizing the *expected violation* in the worst-case scenarios. GrA does not just "cut" edges; it optimizes the explanatory subgraph to minimize this expected tail volatility.
> *   **Adaptability to Diverse Risk Definitions:** Unlike fixed thresholding, which is sensitive to the scale of the metric and often requires manual retuning, CVaR provides a consistent statistical standard (expected tail loss) that adapts to the distribution's shape. This makes our framework **theoretically extensible to other risk proxies** beyond gradient sensitivity—such as gradients of **Mutual Information** objectives or **Entropy $H(\cdot)$**—without requiring heuristic adjustments to the trimming logic.
>
> **2. When gains are limited (Boundary Conditions).**
> We identify scenarios where the benefits of GrA may be more modest:
>
> *   **Flat Risk Distribution:** If the risk distribution is relatively flat (i.e., no pronounced tail), the distinction between the tail mean (CVaR) and the quantile (VaR) diminishes. In such cases, CVaR functionally **converges to a standard percentile rule**, and the specific advantage of our risk-aware formulation is less pronounced.
> *   **Intrinsic Sensitivity:** If an edge lies on a sharp decision boundary (e.g., a critical chemical bond), it is *supposed* to be volatile. Aggressively trimming such "intrinsically high-risk" edges to force stability would incorrectly sacrifice Fidelity. We cannot artificially smooth out explanations if the underlying semantics are inherently steep.
> *   **Violation of Local Linearity:** GrA measures risk via gradients, which is rooted in **first-order sensitivity analysis**. This assumes local linearity. Under massive structural attacks, the graph state moves beyond the "trust region" where this approximation holds. Once high-order curvature dominates, the local gradient is no longer a valid proxy for robustness.
>
> **Action Plan.**
> We will make these distinctions explicit in the revised paper. Specifically, we will:
>
> 1.  Add a discussion in **Section 4.2** connecting the stability gains to the shape of the risk distribution (long-tail vs. flat).
> 2.  **Explicitly clarify the theoretical advantage of CVaR over simple thresholding** in the Methodology section, highlighting its ability to capture tail magnitude and adapt to different metric scales.
> 3.  Expand the **Limitations** section to clarify that GrA targets structurally unstable edges rather than intrinsically sensitive semantics.

---

> > ### Author Response · Authors · 2025-11-28
> > **Summary of Revisions based on your feedback**
> >
> > We thank the reviewer again for the insightful suggestion regarding the method's operating boundaries. To reflect our discussion, we have committed to the following changes in the revised manuscript:
> >
> > 1.  **Clarifying Operating Boundaries (Section 4.2 & Discussion):** We will add a discussion connecting the stability gains to the shape of the empirical risk distribution, explicitly stating that GrA excels in the "Long-tail" regime and may plateau when the distribution is flat.
> > 2.  **Theoretical Contrast (Methodology):** We will explicitly clarify the theoretical advantage of CVaR over simple thresholding (magnitude vs. frequency) to highlight why the method is robust against tail volatility.
> > 3.  **Limitations (Section 6):** We will expand the limitations section to include the "Intrinsic Sensitivity" trade-off and the "Local Linearity" assumption under large perturbation budgets.

---

### Author Response · Authors · 2025-12-04
**Summary of Revisions (Method, Experiments, and Visuals)**

Dear Area Chair and Reviewers,

We have uploaded the revised manuscript. The major revisions concentrate on the **Methodology (Sec 3)** and **Experiments (Sec 4)** sections to address the reviewers' feedback. Key updates include:

**1. Visual & Conceptual Enhancements:**
*   **New Pipeline Figure (Fig. 3):** We completely redrew the framework overview to clearly illustrate the workflow (Relaxation $\to$ Risk Estimation $\to$ Trimming).
*   **New CVaR Illustration (Fig. 2):** We added a conceptual figure to intuitively explain the "Long-tail" risk distribution and demonstrate why CVaR (tail magnitude) is necessary compared to simple thresholding.

**2. Methodological Rigor (Addressing Reviewers gCEu & iDte):**
*   **Surrogate Strategy:** Explicitly formalized the strategy for applying GrA to black-box explainers (Sec 3.4).
*   **Math & Notation:** Corrected the derivation of the **Hutchinson diagonal estimator** and unified all mathematical notation for rigor.
*   **Structure:** Moved dense proofs to **Appendix B** to improve readability.

**3. Experimental Completeness (Addressing Reviewers gCEu & SVTr):**
*   **Consistency:** Completed the evaluation matrix (e.g., adding GNNExplainer on MUTAG) to ensure consistent performance across architectures.

And other changes according to the "Summary of Revisions based on your feedback" in previous discussions.

Best regards,
The Authors

---

### Meta-Review · Area_Chair_7bbk · 2026-01-04

**Summary:**

This paper proposes a training-free framework GrA to improve the stability of post-hoc GNN explainers, which is agnostic to the specific explainer backbone. The main concerns lie in the novelty and the experimental comprehensiveness of GrA. In addition, the manuscript presentation is a weakness as well. According to the author responses and revised manuscript, the concerns about novelty and experimental comprehensiveness remain outstanding. Although GrA is claimed as a model-agnostic framework, the evaluated explainers include merely GNNExplainer and PGExplainer, which cannot sufficiently demonstrate the effectiveness of GrA. Therefore, my overall recommendation is Reject.

**Reviewer Concerns:**

The concerns about manuscript presentation (reviewer iDte, SVTr and gCEu) is roughly addressed in the revised version. However, the novelty (reviewer SVTr and gCEu) and the experimental comprehensiveness (reviewer SVTr and gCEu) are still outstanding.

**Reviewer Scores:**

- **Reviewer LZBa**: Based on the current responses and revision, I think reviewer LZBa won't change his/her score.

- **Reviewer iDte**: The concerns raised by reviewer iDte are mostly open-ended. The author responses are primarily descriptive explanations without substantial supporting evidence. Hence, I think the reviewer iDte won't change the score.

- **Reviewer SVTr** and **gCEu**: Both reviewers question the novelty and experimental comprehensiveness of CVaR. These concerns remain insufficiently addressed after the responses and the revised version. Hence, I think reviewer SVTr and gCEu won't change their score.

---

### Decision · Program_Chairs · 2026-01-26

Reject